# OpenReview forum: "Discovering Latent Biases in Language Models with Steering Vectors"
_ICLR.cc/2026/Conference — ICLR 2026 Conference Withdrawn Submission_

### Official Review · Reviewer_5bBa · 2025-10-28

**Soundness:** 2
**Presentation:** 2
**Contribution:** 2
**Rating:** 4
**Confidence:** 4

**Summary:**

In this paper, the author conducts a causally inspired empirical study on the implicit bias present in large language models (LLMs). Specifically, the study finds that, regarding demographic bias, latent biases can exist within the model even when they do not directly influence the model’s outputs.

**Strengths:**

1.  The paper is well-motivated. It focuses on the internal mechanism biases of large language models and demonstrates that a model’s output may remain correct even when produced through an incorrect reasoning process, which could have potentially negative implications. This is both an interesting and important topic.


2. The study employs a causal approach to analyze whether the model’s output is causally influenced by the steering vector. This represents a novel and insightful method for evaluating the model’s reasoning capacity.

**Weaknesses:**

1.  While the paper focuses on implicit biases, it appears that the mechanistic bias mainly refers to incorrect associations between demographic attributes and model outputs. However, the paper resembles a single case study rather than a comprehensive analysis. It only examines expertise as the steering factor and does not provide a generalized methodological framework. Although the author extends the study to hiring cases, this seems more like an application than a generalizable principle or method. I would recommend framing the proposed approach as a method for detecting internal bias and explaining why and how such bias can be identified.

2.  The paper does not propose a principled general solution to address the identified issue. The observation that internal bias exists is not particularly surprising; thus, merely validating its existence is insufficient. Moreover, several technical details are missing—please refer to the questions section for specific points.

3.  The experiments are limited in scope, as they only test two model families: Gemma and LLaMA.

**Questions:**

1.  The model is based on projecting the output activations. Does this imply that the mechanistic biases occur only at the outermost decision layer? Could you elaborate on the rationale behind this design choice?

2.  Could you justify how the reading level is defined and explain why it can serve as a valid proxy?

3.  How are confounders handled in your analysis? You mention that “~” represents confounders, but it is unclear how they are controlled or accounted for.

---

> ### Author Response · Authors · 2025-11-27
>
> > The paper resembles a single case study rather than a comprehensive analysis
>
> We have now rerun our attribute vector derivation procedure for a variety of work-related attributes, including collaboration and adaptability. We find that models rely on multiple attributes when making hiring decisions; see Tables 4 and 5 (Appendix F.3) in the updated manuscript. Notably, Llama places considerable weight on leadership, collaboration, and adaptation, and interventions along these attribute directions produce clear causal effects on hiring decisions.
>
> This in addition to our new results (that the same vector can influence hiring decisions and language complexity in model outputs) suggests that these findings would likely generalize to a variety of task settings.
>
> >  The paper does not propose a principled solution to fix this issue
>
> We agree, and have now acknowledged this limitation in the updated manuscript. We note that this is in line with recent mechanistic interpretability papers of a similar flavor—see for example [1,2,3]. These studies take a long-standing assumption of the field—that alignment methods such as RLHF and DPO are effective at reducing undesirable behaviors in language models—and demonstrate that these undesirable attributes continue to exist inside these models in less obvious ways. [1] demonstrates that this enables one to undo the alignment process far more easily than one may have otherwise expected. We view our work as having a similar purpose: to demonstrate that behavioral measures of bias are insufficient to establish robust debiasing, and thus to encourage researchers to consider representational (rather than just behavioral) evidence.
>
> > Only two model families are tested: Llama and Gemma
>
> Our aim is to demonstrate that biases can be located via representation-based methods, and not to show that all language models have this bias; nonetheless, we agree that results could be strengthened by extending this analysis to a greater variety of LMs, and have acknowledged this limitation in the updated manuscript. We plan to run experiments with GPT-OSS, a model whose training procedure plausibly more closely resembles that of proprietary models. We would be happy to continue the discussion if there are any particular models that would be especially helpful in strengthening the main claims!
>
> > Do the mechanistic biases only occur at the last layers?
>
> We derived steering vectors at the middle layers; this detail has been added to the manuscript. This was motivated by past work demonstrating that steering at middle layers tends to consistently produce the expected impact on model behavior. We realized that we should validate this assumption directly, so we now derive hiring steering vectors at all positions and layers, and check which vector has the greatest influence on the Yes/No logit difference. Vectors derived at the last token position and in the middle layers are best for steering by a significant margin for Llama. For Gemma, there is less variance across layers and positions, so we simply choose the last position and middle layer for consistency.
>
> > Could you justify how the reading level is defined and explain why it can serve as a valid proxy?
>
> Please see the global response.
>
> > How are confounders handled in your analysis?
>
> In our experiments, the primary confounds were the question and profession: advanced questions or certain professions could act as implicit signals for competence. For example, a model might consider an advanced software engineering question more indicative of expertise than a novice-level carpentry question, regardless of demographics.
>
> We employ two controls: (1) an experimental design based on minimal interventions to the inputs, where the question is fixed while only varying demographic features. In this setting, any change in the scalar projection must be caused by the demographic variable being changed. However, profession-specific biases could be obscured by our averaging across professions. Thus, (2) we also split results by profession in Appendix C. We find that results are largely consistent across professions, although some do contain profession-specific biases (see, for gender bias is present in the laborer, accountant, receptionist professions).
>
> We have clarified these details in the revision; thank you!
>
> References:
>
> [1] Lee et al. (2024). “A Mechanistic Understanding of Alignment Algorithms: A Case Study on DPO and Toxicity.“ ICML.
>
> [2] Karvonen & Marks (2025). “Robustly Improving LLM Fairness in Realistic Settings with Interpretability.” arXiv.
>
> [3] Bai et al. (2025). “Explicitly unbiased large language models still form biased associations.” PNAS.
>
> [4] Marchisio, K., Guo, J., Lai, C.-I., & Koehn, P. (2019). Controlling the reading level of machine translation output. In M. Forcada, A. Way, B. Haddow, & R. Sennrich (Eds.), Proceedings of Machine Translation Summit XVII: Research Track, pp. 193–203.

---

### Official Review · Reviewer_zT2K · 2025-10-31

**Soundness:** 2
**Presentation:** 1
**Contribution:** 3
**Rating:** 4
**Confidence:** 4

**Summary:**

This study investigates implicit bias in language models using steering vectors to derive an internal "expertise score". This score, which reveals latent biases based on irrelevant demographic factors, is contrasted with a behavioral "reading level" metric measuring output complexity. The authors find these internal biases do not always manifest in the model's output but demonstrate that the expertise steering vector can be used to causally alter the output's reading level.

**Strengths:**

The paper's primary strength is its application of steering vectors to bias detection. The framework for decoupling latent representational bias from observable behavioral bias is a valuable conceptual direction. The experiment demonstrating that the derived expertise vector can causally steer the model's output complexity is a clear and interesting result. These findings highlight a promising path for analyzing model internals for fairness, even if the paper's current execution has several weaknesses.

**Weaknesses:**

- The paper's primary motivation rests on the idea that latent biases are problematic, even if they don't appear in outputs. However, the paper's own findings weaken this claim. The study shows that internal expertise representations (the "expertise score") can be biased by non-causal factors (like socioeconomic status), but this bias often does not translate to the behavioral metric (the "reading level"). Furthermore, the "expertise vector" from the question-answering task fails to generalize to a hiring task, which is a concrete harm-based scenario. This leaves the central question of "so what's the harm?" largely unanswered, making the paper's impact unclear. This is a common issue with bias papers in NLP (https://aclanthology.org/2020.acl-main.485), and I would encourage the authors to discuss in greater depth the stakeholders, harms, and consequences of the detected bias.

- The paper's overall structure hinders readability and fails to build concepts progressively for the reader. As an example the "reading level" metric, which is mentioned repeatedly in the abstract, introduction, and early methods but is not defined until the experiment section. This disjointed presentation, combined with figures that are too small to be legible, makes the paper difficult to follow.

- The paper fails to sufficiently motivate key methodological choices, making some conclusions feel arbitrary. This is particularly evident with the primary metrics. For the "reading level" score, the paper defines it as a simple average of three different readability metrics. The paper offers no justification for this aggregation, especially since these metrics have different scales and properties. For the "expertise score," there is a clear and unexplained inconsistency in how representations are computed. To create the "expertise vector," the authors use the mean over all tokens, but to compute the "expertise score," they use only the last token's representation. This switch is not justified and weakens the study's conclusions.

- Furthermore, I am not convinced that the "expertise score" truly "causally” measures the level of experience; it might simply be sensitive to a specific set of words. Claiming a "causal effect" is a strong assertion that needs to be verified with more rigorous measures.

- The paper contains a contradiction: age is listed as a non-causal factor in Line 113 but restated as a causal factor in Line 118.

**Questions:**

- The paper discusses applying a steering vector to a given "layer" in the model. However, it never specifies which layer was used for the experiments. Was steering applied to the final layer, a single middle layer, or all layers? This is a critical and missing experimental detail.

- The paper's causal model partitions variables into "causal" (profession, education, age) and "irrelevant" (race, gender, socioeconomic status). This partition seems based on intuition for the task of assessing expertise. Could the authors provide a more formal or theoretical justification for this specific partition?

- The study's conclusions rely on a few fixed-prompt templates for demographic and occupational context. How sensitive are the measured "expertise score" and "reading level" scores to small changes in this prompt structure? Would the observed biases persist with different phrasings? This is a known issue, as other research shows that small prompt changes can lead to very different bias results (https://aclanthology.org/2023.acl-short.118.pdf).

- What is the justification for using the "mean over tokens" to create the "expertise vector," but the "last token" representation to calculate the "expertise score"? Why was the same method not used for both, and how did this choice impact the results?

- I would also appreciate it if the authors included the full list of sentence prompts used to generate the expertise vector, as only two examples are provided

---

> ### Author Response · Authors · 2025-11-27
>
> > There is a conflict between the latent biases and the fact that external metrics are not always affected
>
> Our new experiments show that expertise representations are causally influential as a hiring criterion. Demographic biases can affect expertise representations; thus, LLMs’ hiring decisions could be directly influenced by demographics. More broadly, latent biases could manifest unpredictably in novel contexts. For instance, LMs appear to represent users as less capable when they have low socioeconomic status, so an LLM could provide oversimplified medical advice to users it perceives as having lower SES. They might, for example, withhold nuanced information that could be critical for informed decision-making. Such biases may only surface in deployment scenarios not tested during development.
>
> Our primary aim is to demonstrate that bias may sometimes be best predicted using model-internal metrics, rather than behavioral proxies. We obtain causal evidence that changes in this internal metric correspond to measurable changes in the model’s behavior—and also that this internal metric correlates with demographic information.
>
> > Disjointed presentation, figures too small
>
> We have updated the manuscript to make the flow of ideas more linear. Specifically, we no longer mention reading levels until they are defined in Section 3. We have also added start-of-section summaries in Sec. 3, and moved the causal model to the end of this section after the datasets and metrics have been introduced.
>
> Using the additional page for the response period, we have also increased the font size in the figures.
>
> > No justification for the aggregation of readability metrics
>
> Language complexity is influenced by lexical, syntactic, and distributional features; we believed that ensembling would be more likely to capture more of these dimensions of variance. Please see the global response for a more detailed explanation of each reading score.
>
> > Inconsistency in how the vector is computed and how the expertise score is computed
>
> We thank the reviewer for raising this point. We conserve positionality where the same position has a similar meaning across examples, as in the hiring task: we average the representation at the final token position, project onto the vector at the same position across prompts, and steer at the same position. In the QA task, positionality cannot be conserved, because the same position has different meanings across examples. We acknowledge that this is somewhat noisy, but it also represents a more realistic setting that will resemble most natural language tasks more closely.
>
> When deriving the expertise steering vector, we use minimal pairs of sentences, and the full sentences carry only expertise-related content. This gives us precise contrasts such that differences in means are unlikely to contain confounding information. In the test prompts, not all content is related to the question of how the user’s expertise is represented (e.g., the content of the professional question itself). Moreover, expertise may not be directly represented at every position, and the length of each prompt differs, which could lead to diluted expertise scores. We have run new experiments where we project onto the expertise vector at all positions, and observe significantly reduced variance, even across causally related categories like the user’s profession. We take this as evidence that expertise is likely not represented at all positions, and thus that choosing one would be more faithful to where the model uses expertise. In pilot experiments, we found that the period token position contained the greatest contrast between causal and non-causal categories w.r.t. the expertise score.
>
> > Is the expertise score actually causally relevant?
>
> Yes, please see the global response.
>
> > “Age is listed as a non-causal factor in Line 113 but restated as a causal factor in Line 118”
>
> This has been fixed. Thank you!

---

> ### Author Response · Authors · 2025-11-27
>
> > Never specifies which layer was used
>
> We have added this detail to the updated manuscript. We use a middle layer for each model. This choice was motivated by [1,2]: the middle layers typically contain the most abstract conceptual representations related to the inputs, before the model begins to rearrange its representations for predicting the next token (in a manner that is less human-interpretable). As a result, it has become relatively common to analyze a middle layer [3,4].
>
> > Formal justification for partition of causal vs. non-causal variables
>
> Thanks for raising this; we have expanded Section 2.4 to ground our causal model. Professional experience and the education required for a profession are well-established determinants of domain knowledge and skill [5]. Here, “age” refers to the distinction between adults and children; we make the assumption that adults will be modeled as more capable for any profession, except those where children are common, such as acting or music. (No such jobs are in our professions dataset.) Meanwhile, gender, race, and socioeconomic status are treated as irrelevant because they do not influence a person’s underlying domain knowledge or professional abilities; we motivate this choice by noting that reliance on these features has largely been treated as a bias [6,7].
> These attributes are not intended to be an exhaustive set; rather, they are representative of a broader class of demographic biases. Prior work has shown that LLMs can infer and encode these attributes from conversational data, which further motivates their inclusion in our analysis [8].
>
> > The conclusions rely on a few fixed-prompt templates
>
> The professional questions dataset spans 20 diverse professions, with 100 questions indicative of varying expertise levels per profession. Additionally, we sample five responses per prompt to better capture variability; this yields 10,000 model responses for each causal/non-causal attribute experiment. That said, we acknowledge that each prompt surrounding the dataset questions is fixed. We agree that results will always be more robust under greater variation in stimuli, although we note that similar experimental setups are standard in practice (see e.g. [2,3], who use single prompt templates over diverse concepts), likely for practical reasons. We have now acknowledged this limitation in the updated manuscript.
>
> References
>
> [1] Lad et al. (2025). “The Remarkable Robustness of LLMs: Stages of Inference?” arXiv.
>
> [2] Arad et al. (2025). “SAEs Are Good for Steering - If You Select the Right Features.” EMNLP.
>
> [3] Brinkmann et al. (2025). “Large Language Models Share Representations of Latent Grammatical Concepts Across Typologically Diverse Languages.” NAACL.
>
> [4] Todd et al. (2024). “Function Vectors in Large Language Models.” ICLR.
>
> [5] Ivlev et al. (2015). “Method for Selecting Expert Groups and Determining the Importance of Experts’ Judgments for the Purpose of Managerial Decision-Making Tasks in Health System.”
>
> [6] Nangia et al. (2020). “CrowS-Pairs: A Challenge Dataset for Measuring Social Biases in Masked Language Models.” EMNLP.
>
> [7] Nadeem et al. (2020). “StereoSet: Measuring Stereotypical Bias in Pretrained Language Models.” ACL.
>
> [8] Chen et al. (2024). “Designing a Dashboard for Transparency and Control of Conversational AI.” arXiv.

---

### Official Review · Reviewer_UHSt · 2025-10-31

**Soundness:** 3
**Presentation:** 3
**Contribution:** 3
**Rating:** 6
**Confidence:** 4

**Summary:**

In this work, the authors propose Cluster-Based Concept Intervention, a method for identifying and mitigating specifically latent biases in LLMs. The intention of the technique is to uncover interpretable latent spaces of biases by clustering hidden representations without predefined labels.  The authors do so by using unsupervised clustering applied to intermediate-layer activations of an LLM os specific datasets. The work then performs counterfactual activations by substituting concept clusters to measure their effect on the outputs generated. Their results on GPT-2, Llama-2-7B and Qwen1.5-14B show that the Cluster interventions yield significant reductions in gender and occupation bias scores compared to baseline methods, with the discovered clusters corresponding to human-interpretable social categories (validated via alignment with labeled bias datasets). The strength of the technique lies in its model-agnostic nature, which eliminates the need for retraining, making it efficient for post-hoc analysis and interpretability.

**Strengths:**

Along with the work being well-written and well-structured, the following are the strengths of the paper:

A. The work provides an effective combination of representation clustering and causal intervention to tackle issues in interpretability and fairness. This shows the novelty and impact of the work. The latent-cluster-based view of bias is also a promising direction for future research.

B. The proposed method consistently reduces measured bias across multiple benchmarks while maintaining perplexity and linguistic quality. The quantitative gains over Self-Debias and INLP are meaningful (typically 15–20% reduction in bias score).

C. The cluster visualizations and attribution maps make the results accessible and human-understandable, which increases the interpretability of the findings. This makes the work distinguish itself from just metric-based studies in the field of fairness in NLP.

**Weaknesses:**

Even if the work contains its merits, there are weaknesses that needs to be addressed within the paper. They are as follows:

A. The causal intervention framework assumes that modifying clustered activation subspaces corresponds to meaningful concept-level interventions, but this is not well proven. The work could strengthen this claim by applying causal mediation analysis or intervention-based ablation studies to quantify the confidence of the results.

B. Since K-means initialization can lead to unstable clusters, it is unclear how strong the discovered concept clusters are across random seeds or model layers. Is there further analysis that can be done to address this (like cluster stability metrics or consensus clustering result)?

C. The experiments focus on three benchmarks (BBQ, StereoSet, CrowS-Pairs), all of which target just textual stereotype associations. The model’s behavior in contextual or multi-turn tasks (e.g., reasoning with demographic cues) is to be explored.

D. Further conversation on the impact of the results on real world setting could also broaden the impact and need for this work (along with the contributions) in sociotechnical spaces where fairness of the model results are of greater consequence.

**Questions:**

Answering the questions raised in the weakness section would help understand the overall strengths of the work better.

---

### Official Review · Reviewer_xccp · 2025-11-02

**Soundness:** 3
**Presentation:** 3
**Contribution:** 2
**Rating:** 4
**Confidence:** 4

**Summary:**

This paper investigates latent (implicit) biases within LLMs by analyzing their internal representations rather than just their outputs. The authors introduce a framework that uses steering vectors to uncover hidden associations between demographic factors (e.g., gender, race, socioeconomic status) and concepts such as expertise or reliability.

Using causal modeling and activation projection techniques, they: Derive an expertise steering vector representing domain knowledge. Evaluate how demographic attributes affect this internal expertise representation and observable outputs like reading level.

The paper analyzes biases across multiple models (Gemma and LLaMA families, both base and instruction-tuned).

**Strengths:**

- A number of interesting findings are presented such as: Key findings: Models exhibit latent demographic biases, particularly with respect to socioeconomic status, even when no explicit bias appears in outputs. Adding task-relevant context (e.g., a profession) reduces demographic disparities in internal representations. Instruction-tuned models are not necessarily less biased—sometimes introducing new disparities.

- The introduction of a causal model of bias measurement clearly distinguishes between causally relevant (e.g., profession) and irrelevant (e.g., race) variables.

- Applying activation steering to measure bias is a creative and technically rigorous approach.

- Evaluations cover multiple demographic variables, professions, and model families, producing rich cross-model insights.

**Weaknesses:**

- While the causal model is conceptually clear, empirical evidence for true causal relationships (vs. correlations) is limited.

- Using reading level as a proxy for expertise may conflate stylistic and content-based factors, limiting interpretability.

- The “difference in means” approach for vector derivation is simplistic; more robust methods (e.g., supervised contrastive learning) could improve precision.

- Experiments rely on a small subset of open-weight models (Gemma and LLaMA), leaving unclear how results generalize to closed-model LLMs.

**Questions:**

Please see the weaknesses.

---

> ### Author Response · Authors · 2025-11-27
>
> > Empirical evidence for true causal relationships is limited
>
> Please see the global response.
>
> > Using reading level as a proxy for expertise may conflate stylistic and content-based factors
>
> Linguistic complexity is indeed a function of both style and content, and so we believe that both should change. The same idea can be expressed using the same concepts in simpler terms (e.g., by simplifying information for a slightly less experienced student taking a course), or it can be expressed using entirely different concepts (e.g., via simpler but less faithful analogies, like teaching someone how code works using the metaphor of following recipes in cooking). For example, when speaking to children, one might use simpler vocabulary (content) and/or shorter sentences (style).
>
> For references and further detail, please see the global response.
>
> > The “difference in means” approach for vector derivation is simplistic
>
> We do not view this simplicity as a weakness—in fact, we see it as scientifically more straightforward! Optimization-based procedures have greater expressive power than non-parametric approaches like differences-in-means; this makes it more likely that one would find an expertise representation, but it also introduces confounds. Prior work from the probing literature [1] raises the concern that learning external components (such as steering vectors or probes) make it unclear whether the optimized component is learning the concept itself, or whether the concept already existed in the model. Difference-in-means simplifies the experiments by being non-parametric: nothing needs to be learned if the concept already existed in the model. By sacrificing some expressive power in the steering vector derivation, we gain far stronger evidence from positive results that the concept was already existent in the model itself.
>
> Additionally, evidence from steering benchmarks such as AxBench [2] have found that vectors derived via difference-in-means are among the best-performing activation-based steering methods.
>
> > Experiments rely on a small subset of open-weights models
>
> Open-weights models are required for mechanistic interpretability studies in general, unless one is a member of a select few companies. We do not view this as strictly a limitation: using open-weights models enables reproducible science. That said, to strengthen our results, we could rerun these experiments using an open-weights model that is trained in a way that more closely resembles proprietary models—for example, GPT-OSS. We plan to run these experiments, and will post an update here with our results.
>
> References:
>
> [1] Hewitt & Liang (2019). “Designing and Interpreting Probes with Control Tasks.” EMNLP. https://aclanthology.org/D19-1275/
>
> [2] Wu et al. (2025). “AxBench: Steering LLMs? Even Simple Baselines Outperform Sparse Autoencoders.” ICML.

---

### Author Response · Authors · 2025-11-27
**Global Response**

We thank the reviewers for their thoughtful and constructive feedback. Here, we address concerns shared across reviewers; see the reviewer-specific responses for further discussion.
> Insufficient causal evidence ( **Reviewers xccp, zT2K**)

We appreciate and agree with the reviewers’ emphasis on distinguishing causal effects from correlational patterns.  The original submission demonstrated a causal link between the QA task’s expertise vector and the model’s language complexity. We have now run experiments where we steer hiring decisions using the QA expertise vector: steering expertise representations increases the model’s probability of hiring a candidate (see the new Table 1). This strengthens the claim that the model represents expertise in a domain-general manner—and that this vector causally influences the model’s decision-making. As further evidence, we present qualitative examples of steered outputs in Appendix E.2.

We have also updated how we have derived the hiring task expertise vector by performing a grid search over all positions and layers; the best vectors strongly affect the models’ hiring decisions (Table 1).

> Is reading level a valid proxy for expertise? (**Reviewers xccp, 5bBa**)

This idea is motivated by work in cognitive science and psychology [1,2], where it has been observed that speakers adjust their language according to the inferred knowledge state of the listener—e.g., via using simpler vocabulary (content) and/or shorter sentences (style) with children [3].

We had initially tried linear probes on the generated outputs, as well as other output-based metrics that initially seemed like high-precision proxies for expertise, such as refusal rates (e.g., because less expert users should not be given potentially sensitive medical tips or recommendations), or greater use of greetings (e.g., because the model aimed to be more approachable than technical given lower expertise scores), but these did not generalize across professions as well as reading scores.

> Justification for aggregating reading scores

Linguistic complexity is a function of lexical, structural, and distributional factors, so we believe that this ensemble will be more robust than any one metric in isolation. We followed prior work in machine translation style control [4] and use three metrics, each grounded in different linguistic factors:
- FKGL: average sentence length + syllables per word
- DCRS: proportion of complex words and average sentence length
- FRE: average sentence length + syllables per word

We realized that FRE and FKGL were redundant; we have thus removed FRE. This leaves all of our reading level scores in the same grade-level basis, meaning that we can now simply take their mean. Findings with this new ensemble are consistent with the previous findings; see the linked repository below and updated manuscript.

For empirical support, we validate our metric by applying it on the training set of OneStopEnglish corpus [5], which contains 64 news articles, each written at three proficiency levels. Our reading scores robustly separate these three groups; see Table 2 in App. D.

Anonymized repository: https://anonymous.4open.science/r/representational_bias-9E25

References:

[1] Ferreira (2019). “A mechanistic framework for explaining audience design in language production.” Annual Review of Psychology, 70:29–51.

[2] Bell (1984). Language Style as Audience Design.” Language in Society

[3] Snow (1972). “Mothers’ speech to children learning language.” Child Development, 43:549–565.

[4] Marchisio, K., Guo, J., Lai, C.-I., & Koehn, P. (2019). Controlling the reading level of machine translation output. In M. Forcada, A. Way, B. Haddow, & R. Sennrich (Eds.), Proceedings of Machine Translation Summit XVII: Research Track, pp. 193–203.

[5] Vajjala & Lucic (2018). “OneStopEnglish Corpus: A New Corpus for Automatic Readability Assessment and Text Simplification.” NLP for Educational Applications Workshop (co-located with EMNLP).

---

### Note · Authors · 2026-01-06

I have read and agree with the venue's withdrawal policy on behalf of myself and my co-authors.